# Attention Is All You Need


**Ashish Vaswani**[*]
Google Brain
avaswani@google.com

**Noam Shazeer**[*]
Google Brain
noam@google.com

**Niki Parmar**[*]
Google Research
nikip@google.com

**Jakob Uszkoreit**[*]
Google Research
usz@google.com

**Llion Jones**[*]
Google Research
llion@google.com

**Aidan N. Gomez**[*] [†]
University of Toronto
aidan@cs.toronto.edu

**Łukasz Kaiser**[*]
Google Brain
lukaszkaiser@google.com

**Illia Polosukhin**[*] [‡]
illia.polosukhin@gmail.com



## Abstract

The dominant sequence transduction models are based on complex recurrent or convolutional neural networks that include an encoder and a decoder. The best performing models also connect the encoder and decoder through an attention mechanism. We propose a new simple network architecture, the Transformer, based solely on attention mechanisms, dispensing with recurrence and convolutions entirely. Experiments on two machine translation tasks show these models to be superior in quality while being more parallelizable and requiring significantly less time to train. Our model achieves 28.4 BLEU on the WMT 2014 English-to-German translation task, improving over the existing best results, including ensembles, by over 2 BLEU. On the WMT 2014 English-to-French translation task, our model establishes a new single-model state-of-the-art BLEU score of 41.8 after training for 3.5 days on eight GPUs, a small fraction of the training costs of the best models from the literature. We show that the Transformer generalizes well to other tasks by applying it successfully to English constituency parsing both with large and limited training data.


## 1 Introduction

Recurrent neural networks, long short-term memory [13] and gated recurrent [7] neural networks in particular, have been firmly established as state of the art approaches in sequence modeling and

---

[*]Equal contribution. Listing order is random. Jakob proposed replacing RNNs with self-attention and started the effort to evaluate this idea. Ashish, with Illia, designed and implemented the first Transformer models and has been crucially involved in every aspect of this work. Noam proposed scaled dot-product attention, multi-head attention and the parameter-free position representation and became the other person involved in nearly every detail. Niki designed, implemented, tuned and evaluated countless model variants in our original codebase and tensor2tensor. Llion also experimented with novel model variants, was responsible for our initial codebase, and efficient inference and visualizations. Lukasz and Aidan spent countless long days designing various parts of and implementing tensor2tensor, replacing our earlier codebase, greatly improving results and massively accelerating our research.

[†]Work performed while at Google Brain.

[‡]Work performed while at Google Research.



transduction problems such as language modeling and machine translation [35, 2, 5]. Numerous efforts have since continued to push the boundaries of recurrent language models and encoder-decoder architectures [38, 24, 15].

Recurrent models typically factor computation along the symbol positions of the input and output sequences. Aligning the positions to steps in computation time, they generate a sequence of hidden states $h_t$, as a function of the previous hidden state $h_{t-1}$ and the input for position $t$. This inherently sequential nature precludes parallelization within training examples, which becomes critical at longer sequence lengths, as memory constraints limit batching across examples. Recent work has achieved significant improvements in computational efficiency through factorization tricks [21] and conditional computation [32], while also improving model performance in case of the latter. The fundamental constraint of sequential computation, however, remains.

Attention mechanisms have become an integral part of compelling sequence modeling and transduction models in various tasks, allowing modeling of dependencies without regard to their distance in the input or output sequences [2, 19]. In all but a few cases [27], however, such attention mechanisms are used in conjunction with a recurrent network.

In this work we propose the Transformer, a model architecture eschewing recurrence and instead relying entirely on an attention mechanism to draw global dependencies between input and output. The Transformer allows for significantly more parallelization and can reach a new state of the art in translation quality after being trained for as little as twelve hours on eight P100 GPUs.

## 2 Background

The goal of reducing sequential computation also forms the foundation of the Extended Neural GPU [16], ByteNet [18] and ConvS2S [9], all of which use convolutional neural networks as basic building block, computing hidden representations in parallel for all input and output positions. In these models, the number of operations required to relate signals from two arbitrary input or output positions grows in the distance between positions, linearly for ConvS2S and logarithmically for ByteNet. This makes it more difficult to learn dependencies between distant positions [12]. In the Transformer this is reduced to a constant number of operations, albeit at the cost of reduced effective resolution due to averaging attention-weighted positions, an effect we counteract with Multi-Head Attention as described in section 3.2.

Self-attention, sometimes called intra-attention is an attention mechanism relating different positions of a single sequence in order to compute a representation of the sequence. Self-attention has been used successfully in a variety of tasks including reading comprehension, abstractive summarization, textual entailment and learning task-independent sentence representations [4, 27, 28, 22].

End-to-end memory networks are based on a recurrent attention mechanism instead of sequence-aligned recurrence and have been shown to perform well on simple-language question answering and language modeling tasks [34].

To the best of our knowledge, however, the Transformer is the first transduction model relying entirely on self-attention to compute representations of its input and output without using sequence-aligned RNNs or convolution. In the following sections, we will describe the Transformer, motivate self-attention and discuss its advantages over models such as [17, 18] and [9].

## 3 Model Architecture

Most competitive neural sequence transduction models have an encoder-decoder structure [5, 2, 35]. Here, the encoder maps an input sequence of symbol representations $(x_1, ..., x_n)$ to a sequence of continuous representations $\mathbf{z} = (z_1, ..., z_n)$. Given $\mathbf{z}$, the decoder then generates an output sequence $(y_1, ..., y_m)$ of symbols one element at a time. At each step the model is auto-regressive [10], consuming the previously generated symbols as additional input when generating the next.

The Transformer follows this overall architecture using stacked self-attention and point-wise, fully connected layers for both the encoder and decoder, shown in the left and right halves of Figure 1, respectively.



Figure 1: The Transformer - model architecture.

### 3.1 Encoder and Decoder Stacks

**Encoder:** The encoder is composed of a stack of $N = 6$ identical layers. Each layer has two sub-layers. The first is a multi-head self-attention mechanism, and the second is a simple, position-wise fully connected feed-forward network. We employ a residual connection [11] around each of the two sub-layers, followed by layer normalization [1]. That is, the output of each sub-layer is $\text{LayerNorm}(x + \text{Sublayer}(x))$, where $\text{Sublayer}(x)$ is the function implemented by the sub-layer itself. To facilitate these residual connections, all sub-layers in the model, as well as the embedding layers, produce outputs of dimension $d_{\text{model}} = 512$.

**Decoder:** The decoder is also composed of a stack of $N = 6$ identical layers. In addition to the two sub-layers in each encoder layer, the decoder inserts a third sub-layer, which performs multi-head attention over the output of the encoder stack. Similar to the encoder, we employ residual connections around each of the sub-layers, followed by layer normalization. We also modify the self-attention sub-layer in the decoder stack to prevent positions from attending to subsequent positions. This masking, combined with fact that the output embeddings are offset by one position, ensures that the predictions for position $i$ can depend only on the known outputs at positions less than $i$.

### 3.2 Attention

An attention function can be described as mapping a query and a set of key-value pairs to an output, where the query, keys, values, and output are all vectors. The output is computed as a weighted sum of the values, where the weight assigned to each value is computed by a compatibility function of the query with the corresponding key.



Figure 2: (left) Scaled Dot-Product Attention. (right) Multi-Head Attention consists of several attention layers running in parallel.

### 3.2.1 Scaled Dot-Product Attention

We call our particular attention "Scaled Dot-Product Attention" (Figure 2). The input consists of queries and keys of dimension $d_k$, and values of dimension $d_v$. We compute the dot products of the query with all keys, divide each by $\sqrt{d_k}$, and apply a softmax function to obtain the weights on the values.

In practice, we compute the attention function on a set of queries simultaneously, packed together into a matrix $Q$. The keys and values are also packed together into matrices $K$ and $V$. We compute the matrix of outputs as:

$$\text{Attention}(Q, K, V) = \text{softmax}(\frac{QK^T}{\sqrt{d_k}})V \tag{1}$$

The two most commonly used attention functions are additive attention [2], and dot-product (multiplicative) attention. Dot-product attention is identical to our algorithm, except for the scaling factor of $\frac{1}{\sqrt{d_k}}$. Additive attention computes the compatibility function using a feed-forward network with a single hidden layer. While the two are similar in theoretical complexity, dot-product attention is much faster and more space-efficient in practice, since it can be implemented using highly optimized matrix multiplication code.

While for small values of $d_k$ the two mechanisms perform similarly, additive attention outperforms dot product attention without scaling for larger values of $d_k$ [3]. We suspect that for large values of $d_k$, the dot products grow large in magnitude, pushing the softmax function into regions where it has extremely small gradients [4]. To counteract this effect, we scale the dot products by $\frac{1}{\sqrt{d_k}}$.

### 3.2.2 Multi-Head Attention

Instead of performing a single attention function with $d_{\text{model}}$-dimensional keys, values and queries, we found it beneficial to linearly project the queries, keys and values $h$ times with different, learned linear projections to $d_k$, $d_k$ and $d_v$ dimensions, respectively. On each of these projected versions of queries, keys and values we then perform the attention function in parallel, yielding $d_v$-dimensional output values. These are concatenated and once again projected, resulting in the final values, as depicted in Figure 2.

---

[4]To illustrate why the dot products get large, assume that the components of $q$ and $k$ are independent random variables with mean 0 and variance 1. Then their dot product, $q \cdot k = \sum_{i=1}^{d_k} q_i k_i$, has mean 0 and variance $d_k$.

4

Multi-head attention allows the model to jointly attend to information from different representation subspaces at different positions. With a single attention head, averaging inhibits this.

$$\text{MultiHead}(Q, K, V) = \text{Concat}(\text{head}_1, ..., \text{head}_h)W^O$$
$$\text{where head}_i = \text{Attention}(QW_i^Q, KW_i^K, VW_i^V)$$

Where the projections are parameter matrices $W_i^Q \in \mathbb{R}^{d_{\text{model}} \times d_k}$, $W_i^K \in \mathbb{R}^{d_{\text{model}} \times d_k}$, $W_i^V \in \mathbb{R}^{d_{\text{model}} \times d_v}$ and $W^O \in \mathbb{R}^{hd_v \times d_{\text{model}}}$.

In this work we employ $h = 8$ parallel attention layers, or heads. For each of these we use $d_k = d_v = d_{\text{model}}/h = 64$. Due to the reduced dimension of each head, the total computational cost is similar to that of single-head attention with full dimensionality.

### 3.2.3 Applications of Attention in our Model

The Transformer uses multi-head attention in three different ways:

- In "encoder-decoder attention" layers, the queries come from the previous decoder layer, and the memory keys and values come from the output of the encoder. This allows every position in the decoder to attend over all positions in the input sequence. This mimics the typical encoder-decoder attention mechanisms in sequence-to-sequence models such as [38, 2, 9].
- The encoder contains self-attention layers. In a self-attention layer all of the keys, values and queries come from the same place, in this case, the output of the previous layer in the encoder. Each position in the encoder can attend to all positions in the previous layer of the encoder.
- Similarly, self-attention layers in the decoder allow each position in the decoder to attend to all positions in the decoder up to and including that position. We need to prevent leftward information flow in the decoder to preserve the auto-regressive property. We implement this inside of scaled dot-product attention by masking out (setting to $-\infty$) all values in the input of the softmax which correspond to illegal connections. See Figure 2.

## 3.3 Position-wise Feed-Forward Networks

In addition to attention sub-layers, each of the layers in our encoder and decoder contains a fully connected feed-forward network, which is applied to each position separately and identically. This consists of two linear transformations with a ReLU activation in between.

$$\text{FFN}(x) = \max(0, xW_1 + b_1)W_2 + b_2 \tag{2}$$

While the linear transformations are the same across different positions, they use different parameters from layer to layer. Another way of describing this is as two convolutions with kernel size 1. The dimensionality of input and output is $d_{\text{model}} = 512$, and the inner-layer has dimensionality $d_{ff} = 2048$.

## 3.4 Embeddings and Softmax

Similarly to other sequence transduction models, we use learned embeddings to convert the input tokens and output tokens to vectors of dimension $d_{\text{model}}$. We also use the usual learned linear transformation and softmax function to convert the decoder output to predicted next-token probabilities. In our model, we share the same weight matrix between the two embedding layers and the pre-softmax linear transformation, similar to [30]. In the embedding layers, we multiply those weights by $\sqrt{d_{\text{model}}}$.

## 3.5 Positional Encoding

Since our model contains no recurrence and no convolution, in order for the model to make use of the order of the sequence, we must inject some information about the relative or absolute position of the



Table 1: Maximum path lengths, per-layer complexity and minimum number of sequential operations for different layer types. $n$ is the sequence length, $d$ is the representation dimension, $k$ is the kernel size of convolutions and $r$ the size of the neighborhood in restricted self-attention.

| Layer Type | Complexity per Layer | Sequential Operations | Maximum Path Length |
| --- | --- | --- | --- |
| Self-Attention | $O(n^2 \cdot d)$ | $O(1)$ | $O(1)$ |
| Recurrent | $O(n \cdot d^2)$ | $O(n)$ | $O(n)$ |
| Convolutional | $O(k \cdot n \cdot d^2)$ | $O(1)$ | $O(log_k(n))$ |
| Self-Attention (restricted) | $O(r \cdot n \cdot d)$ | $O(1)$ | $O(n/r)$ |

tokens in the sequence. To this end, we add "positional encodings" to the input embeddings at the bottoms of the encoder and decoder stacks. The positional encodings have the same dimension $d_{\text{model}}$ as the embeddings, so that the two can be summed. There are many choices of positional encodings, learned and fixed [9].

In this work, we use sine and cosine functions of different frequencies:

$$PE_{(pos,2i)} = sin(pos/10000^{2i/d_{\text{model}}})$$
$$PE_{(pos,2i+1)} = cos(pos/10000^{2i/d_{\text{model}}})$$

where $pos$ is the position and $i$ is the dimension. That is, each dimension of the positional encoding corresponds to a sinusoid. The wavelengths form a geometric progression from $2\pi$ to $10000 \cdot 2\pi$. We chose this function because we hypothesized it would allow the model to easily learn to attend by relative positions, since for any fixed offset $k$, $PE_{pos+k}$ can be represented as a linear function of $PE_{pos}$.

We also experimented with using learned positional embeddings [9] instead, and found that the two versions produced nearly identical results (see Table 3 row (E)). We chose the sinusoidal version because it may allow the model to extrapolate to sequence lengths longer than the ones encountered during training.

## 4 Why Self-Attention

In this section we compare various aspects of self-attention layers to the recurrent and convolutional layers commonly used for mapping one variable-length sequence of symbol representations $(x_1, ..., x_n)$ to another sequence of equal length $(z_1, ..., z_n)$, with $x_i, z_i \in \mathbb{R}^d$, such as a hidden layer in a typical sequence transduction encoder or decoder. Motivating our use of self-attention we consider three desiderata.

One is the total computational complexity per layer. Another is the amount of computation that can be parallelized, as measured by the minimum number of sequential operations required.

The third is the path length between long-range dependencies in the network. Learning long-range dependencies is a key challenge in many sequence transduction tasks. One key factor affecting the ability to learn such dependencies is the length of the paths forward and backward signals have to traverse in the network. The shorter these paths between any combination of positions in the input and output sequences, the easier it is to learn long-range dependencies [12]. Hence we also compare the maximum path length between any two input and output positions in networks composed of the different layer types.

As noted in Table 1, a self-attention layer connects all positions with a constant number of sequentially executed operations, whereas a recurrent layer requires $O(n)$ sequential operations. In terms of computational complexity, self-attention layers are faster than recurrent layers when the sequence length $n$ is smaller than the representation dimensionality $d$, which is most often the case with sentence representations used by state-of-the-art models in machine translations, such as word-piece [38] and byte-pair [31] representations. To improve computational performance for tasks involving very long sequences, self-attention could be restricted to considering only a neighborhood of size $r$ in



the input sequence centered around the respective output position. This would increase the maximum path length to $O(n/r)$. We plan to investigate this approach further in future work.

A single convolutional layer with kernel width $k < n$ does not connect all pairs of input and output positions. Doing so requires a stack of $O(n/k)$ convolutional layers in the case of contiguous kernels, or $O(log_k(n))$ in the case of dilated convolutions [18], increasing the length of the longest paths between any two positions in the network. Convolutional layers are generally more expensive than recurrent layers, by a factor of $k$. Separable convolutions [6], however, decrease the complexity considerably, to $O(k \cdot n \cdot d + n \cdot d^2)$. Even with $k = n$, however, the complexity of a separable convolution is equal to the combination of a self-attention layer and a point-wise feed-forward layer, the approach we take in our model.

As side benefit, self-attention could yield more interpretable models. We inspect attention distributions from our models and present and discuss examples in the appendix. Not only do individual attention heads clearly learn to perform different tasks, many appear to exhibit behavior related to the syntactic and semantic structure of the sentences.

## 5 Training

This section describes the training regime for our models.

### 5.1 Training Data and Batching

We trained on the standard WMT 2014 English-German dataset consisting of about 4.5 million sentence pairs. Sentences were encoded using byte-pair encoding [3], which has a shared source-target vocabulary of about 37000 tokens. For English-French, we used the significantly larger WMT 2014 English-French dataset consisting of 36M sentences and split tokens into a 32000 word-piece vocabulary [38]. Sentence pairs were batched together by approximate sequence length. Each training batch contained a set of sentence pairs containing approximately 25000 source tokens and 25000 target tokens.

### 5.2 Hardware and Schedule

We trained our models on one machine with 8 NVIDIA P100 GPUs. For our base models using the hyperparameters described throughout the paper, each training step took about 0.4 seconds. We trained the base models for a total of 100,000 steps or 12 hours. For our big models,(described on the bottom line of table 3), step time was 1.0 seconds. The big models were trained for 300,000 steps (3.5 days).

### 5.3 Optimizer

We used the Adam optimizer [20] with $\beta_1 = 0.9$, $\beta_2 = 0.98$ and $\epsilon = 10^{-9}$. We varied the learning rate over the course of training, according to the formula:

$$lrate = d_{\text{model}}^{-0.5} \cdot \min(step\_num^{-0.5}, step\_num \cdot warmup\_steps^{-1.5}) \qquad (3)$$

This corresponds to increasing the learning rate linearly for the first $warmup\_steps$ training steps, and decreasing it thereafter proportionally to the inverse square root of the step number. We used $warmup\_steps = 4000$.

### 5.4 Regularization

We employ three types of regularization during training:

**Residual Dropout** We apply dropout [33] to the output of each sub-layer, before it is added to the sub-layer input and normalized. In addition, we apply dropout to the sums of the embeddings and the positional encodings in both the encoder and decoder stacks. For the base model, we use a rate of $P_{drop} = 0.1$.



Table 2: The Transformer achieves better BLEU scores than previous state-of-the-art models on the English-to-German and English-to-French newstest2014 tests at a fraction of the training cost.

| Model | BLEU | | Training Cost (FLOPs) | |
|---|---|---|---|---|
| | EN-DE | EN-FR | EN-DE | EN-FR |
| ByteNet [18] | 23.75 | | | |
| Deep-Att + PosUnk [39] | | 39.2 | | $1.0 \cdot 10^{20}$ |
| GNMT + RL [38] | 24.6 | 39.92 | $2.3 \cdot 10^{19}$ | $1.4 \cdot 10^{20}$ |
| ConvS2S [9] | 25.16 | 40.46 | $9.6 \cdot 10^{18}$ | $1.5 \cdot 10^{20}$ |
| MoE [32] | 26.03 | 40.56 | $2.0 \cdot 10^{19}$ | $1.2 \cdot 10^{20}$ |
| Deep-Att + PosUnk Ensemble [39] | | 40.4 | | $8.0 \cdot 10^{20}$ |
| GNMT + RL Ensemble [38] | 26.30 | 41.16 | $1.8 \cdot 10^{20}$ | $1.1 \cdot 10^{21}$ |
| ConvS2S Ensemble [9] | 26.36 | **41.29** | $7.7 \cdot 10^{19}$ | $1.2 \cdot 10^{21}$ |
| Transformer (base model) | 27.3 | 38.1 | $\mathbf{3.3 \cdot 10^{18}}$ | |
| Transformer (big) | **28.4** | **41.8** | $2.3 \cdot 10^{19}$ | |

**Label Smoothing** During training, we employed label smoothing of value $\epsilon_{ls} = 0.1$ [36]. This hurts perplexity, as the model learns to be more unsure, but improves accuracy and BLEU score.

## 6 Results

### 6.1 Machine Translation

On the WMT 2014 English-to-German translation task, the big transformer model (Transformer (big) in Table 2) outperforms the best previously reported models (including ensembles) by more than $2.0$ BLEU, establishing a new state-of-the-art BLEU score of $28.4$. The configuration of this model is listed in the bottom line of Table 3. Training took $3.5$ days on $8$ P100 GPUs. Even our base model surpasses all previously published models and ensembles, at a fraction of the training cost of any of the competitive models.

On the WMT 2014 English-to-French translation task, our big model achieves a BLEU score of $41.0$, outperforming all of the previously published single models, at less than $1/4$ the training cost of the previous state-of-the-art model. The Transformer (big) model trained for English-to-French used dropout rate $P_{drop} = 0.1$, instead of $0.3$.

For the base models, we used a single model obtained by averaging the last 5 checkpoints, which were written at 10-minute intervals. For the big models, we averaged the last 20 checkpoints. We used beam search with a beam size of 4 and length penalty $\alpha = 0.6$ [38]. These hyperparameters were chosen after experimentation on the development set. We set the maximum output length during inference to input length + $50$, but terminate early when possible [38].

Table 2 summarizes our results and compares our translation quality and training costs to other model architectures from the literature. We estimate the number of floating point operations used to train a model by multiplying the training time, the number of GPUs used, and an estimate of the sustained single-precision floating-point capacity of each GPU [5].

### 6.2 Model Variations

To evaluate the importance of different components of the Transformer, we varied our base model in different ways, measuring the change in performance on English-to-German translation on the development set, newstest2013. We used beam search as described in the previous section, but no checkpoint averaging. We present these results in Table 3.

In Table 3 rows (A), we vary the number of attention heads and the attention key and value dimensions, keeping the amount of computation constant, as described in Section 3.2.2. While single-head attention is 0.9 BLEU worse than the best setting, quality also drops off with too many heads.

---
[5]We used values of 2.8, 3.7, 6.0 and 9.5 TFLOPS for K80, K40, M40 and P100, respectively.



Table 3: Variations on the Transformer architecture. Unlisted values are identical to those of the base model. All metrics are on the English-to-German translation development set, newstest2013. Listed perplexities are per-wordpiece, according to our byte-pair encoding, and should not be compared to per-word perplexities.

|   | $N$ | $d_{\text{model}}$ | $d_{\text{ff}}$ | $h$ | $d_k$ | $d_v$ | $P_{drop}$ | $\epsilon_{ls}$ | train steps | PPL (dev) | BLEU (dev) | params $\times 10^6$ |
|---|---|---|---|---|---|---|---|---|---|---|---|---|
| base | 6 | 512 | 2048 | 8 | 64 | 64 | 0.1 | 0.1 | 100K | 4.92 | 25.8 | 65 |
| (A) |   |   |   | 1 | 512 | 512 |   |   |   | 5.29 | 24.9 |   |
|     |   |   |   | 4 | 128 | 128 |   |   |   | 5.00 | 25.5 |   |
|     |   |   |   | 16 | 32 | 32 |   |   |   | 4.91 | 25.8 |   |
|     |   |   |   | 32 | 16 | 16 |   |   |   | 5.01 | 25.4 |   |
| (B) |   |   |   |   | 16 |   |   |   |   | 5.16 | 25.1 | 58 |
|     |   |   |   |   | 32 |   |   |   |   | 5.01 | 25.4 | 60 |
| (C) | 2 |   |   |   |   |   |   |   |   | 6.11 | 23.7 | 36 |
|     | 4 |   |   |   |   |   |   |   |   | 5.19 | 25.3 | 50 |
|     | 8 |   |   |   |   |   |   |   |   | 4.88 | 25.5 | 80 |
|     |   | 256 |   |   | 32 | 32 |   |   |   | 5.75 | 24.5 | 28 |
|     |   | 1024 |   |   | 128 | 128 |   |   |   | 4.66 | 26.0 | 168 |
|     |   |   | 1024 |   |   |   |   |   |   | 5.12 | 25.4 | 53 |
|     |   |   | 4096 |   |   |   |   |   |   | 4.75 | 26.2 | 90 |
| (D) |   |   |   |   |   |   | 0.0 |   |   | 5.77 | 24.6 |   |
|     |   |   |   |   |   |   | 0.2 |   |   | 4.95 | 25.5 |   |
|     |   |   |   |   |   |   |   | 0.0 |   | 4.67 | 25.3 |   |
|     |   |   |   |   |   |   |   | 0.2 |   | 5.47 | 25.7 |   |
| (E) |   | positional embedding instead of sinusoids |   |   |   |   |   |   |   | 4.92 | 25.7 |   |
| big | 6 | 1024 | 4096 | 16 |   |   | 0.3 |   | 300K | **4.33** | **26.4** | 213 |

Table 4: The Transformer generalizes well to English constituency parsing (Results are on Section 23 of WSJ)

| Parser | Training | WSJ 23 F1 |
|---|---|---|
| Vinyals & Kaiser el al. (2014) [37] | WSJ only, discriminative | 88.3 |
| Petrov et al. (2006) [29] | WSJ only, discriminative | 90.4 |
| Zhu et al. (2013) [40] | WSJ only, discriminative | 90.4 |
| Dyer et al. (2016) [8] | WSJ only, discriminative | 91.7 |
| Transformer (4 layers) | WSJ only, discriminative | 91.3 |
| Zhu et al. (2013) [40] | semi-supervised | 91.3 |
| Huang & Harper (2009) [14] | semi-supervised | 91.3 |
| McClosky et al. (2006) [26] | semi-supervised | 92.1 |
| Vinyals & Kaiser el al. (2014) [37] | semi-supervised | 92.1 |
| Transformer (4 layers) | semi-supervised | 92.7 |
| Luong et al. (2015) [23] | multi-task | 93.0 |
| Dyer et al. (2016) [8] | generative | 93.3 |

In Table 3 rows (B), we observe that reducing the attention key size $d_k$ hurts model quality. This suggests that determining compatibility is not easy and that a more sophisticated compatibility function than dot product may be beneficial. We further observe in rows (C) and (D) that, as expected, bigger models are better, and dropout is very helpful in avoiding over-fitting. In row (E) we replace our sinusoidal positional encoding with learned positional embeddings [9], and observe nearly identical results to the base model.

### 6.3 English Constituency Parsing

To evaluate if the Transformer can generalize to other tasks we performed experiments on English constituency parsing. This task presents specific challenges: the output is subject to strong structural

9

constraints and is significantly longer than the input. Furthermore, RNN sequence-to-sequence models have not been able to attain state-of-the-art results in small-data regimes [37].

We trained a 4-layer transformer with $d_{model} = 1024$ on the Wall Street Journal (WSJ) portion of the Penn Treebank [25], about 40K training sentences. We also trained it in a semi-supervised setting, using the larger high-confidence and BerkleyParser corpora from with approximately 17M sentences [37]. We used a vocabulary of 16K tokens for the WSJ only setting and a vocabulary of 32K tokens for the semi-supervised setting.

We performed only a small number of experiments to select the dropout, both attention and residual (section 5.4), learning rates and beam size on the Section 22 development set, all other parameters remained unchanged from the English-to-German base translation model. During inference, we increased the maximum output length to input length + 300. We used a beam size of 21 and $\alpha = 0.3$ for both WSJ only and the semi-supervised setting.

Our results in Table 4 show that despite the lack of task-specific tuning our model performs surprisingly well, yielding better results than all previously reported models with the exception of the Recurrent Neural Network Grammar [8].

In contrast to RNN sequence-to-sequence models [37], the Transformer outperforms the BerkeleyParser [29] even when training only on the WSJ training set of 40K sentences.

## 7 Conclusion

In this work, we presented the Transformer, the first sequence transduction model based entirely on attention, replacing the recurrent layers most commonly used in encoder-decoder architectures with multi-headed self-attention.

For translation tasks, the Transformer can be trained significantly faster than architectures based on recurrent or convolutional layers. On both WMT 2014 English-to-German and WMT 2014 English-to-French translation tasks, we achieve a new state of the art. In the former task our best model outperforms even all previously reported ensembles.

We are excited about the future of attention-based models and plan to apply them to other tasks. We plan to extend the Transformer to problems involving input and output modalities other than text and to investigate local, restricted attention mechanisms to efficiently handle large inputs and outputs such as images, audio and video. Making generation less sequential is another research goals of ours.

The code we used to train and evaluate our models is available at https://github.com/tensorflow/tensor2tensor.

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

# Attention Visualizations

Figure 3: An example of the attention mechanism following long-distance dependencies in the encoder self-attention in layer 5 of 6. Many of the attention heads attend to a distant dependency of the verb 'making', completing the phrase 'making...more difficult'. Attentions here shown only for the word 'making'. Different colors represent different heads. Best viewed in color.

13

Figure 4: Two attention heads, also in layer 5 of 6, apparently involved in anaphora resolution. Top: Full attentions for head 5. Bottom: Isolated attentions from just the word 'its' for attention heads 5 and 6. Note that the attentions are very sharp for this word.

14

Figure 5: Many of the attention heads exhibit behaviour that seems related to the structure of the sentence. We give two such examples above, from two different heads from the encoder self-attention at layer 5 of 6. The heads clearly learned to perform different tasks.

15