[Reviews · NeurIPS 2017]

Reviewer 1



This work introduces a quite strikingly different approach to the problem of sequence-to-sequence modeling, by utilizing several different layers of self-attention combined with a standard attention. The work uses a variant of dot-product attention with multiple heads that can both be computed very quickly (particularly on GPU). When combined with temporal embeddings, layer norm, and several other tricks, this technique can replace the use of RNNs entirely on seq2seq models. Since this removes a serial training bottleneck, the whole system can be trained much more efficiently. Even better the system achieves state-of-the-art results on translation, and significantly improves the performance of seq2seq only parsing models. I feel this work is a clear accept. Seq2seq is so influential that major improvements of this form will have significant impact on the field of NLP almost instantly. This work is already the talk of the community, and many people are trying to replicate these results already. While none of the underlying techniques here are strikingly novel in themselves, the combination of them and the details necessary for getting it to work as well as LSTMs is a major achievement. As part of this review, I spent a lot of time reimplementing the work and looking through the code. Here are a couple suggestions of areas that I got tripped up on: - There are a lot of hyperparameters in the code itself that I had to extract, might be nice to include these in the paper. - The learning rate schedule seems to really matter. Using simple SGD works fine for LSTM, but seems to fail here - Inference for this problem is quite different than other NMT systems, might be worth discussing a bit more.

Reviewer 2



The paper presents a new architecture for encoder/decoder models for sequence-to-sequence modeling that is solely based on (multi-layered) attention networks combined with standard Feed-Forward networks as opposed to the common scheme of using recurrent or convolutional neural networks. The paper presents two main advantages of this new architecture: (1) Reduced training time due to reduced complexity of the architecture, and (2) new State-of-the-Art result on standard WMT data sets, outperforming previous work by about 1 BLEU point. Strengths: - The paper argues well that (1) can be achieved by avoiding recurrent or convolutional layers and the complexity analysis in Table 1 strengthens the argument. - (2) is shown by comparing the model performance against strong baselines on two language pairs, English-German and English-French. The main strengths of the paper are that it proposes an entirely novel architecture without recurrence or convolutions, and advances state of the art. Weaknesses: - While the general architecture of the model is described well and is illustrated by figures, architectural details lack mathematical definition, for example multi-head attention. Why is there a split arrow in Figure 2 right, bottom right? I assume these are the inputs for the attention layer, namely query, keys, and values. Are the same vectors used for keys and values here or different sections of them? A formal definition of this would greatly help readers understand this. - The proposed model contains lots of hyperparameters, and the most important ones are evaluated in ablation studies in the experimental section. It would have been nice to see significance tests for the various configurations in Table 3. - The complexity argument claims that self-attention models have a maximum path length of 1 which should help maintaining information flow between distant symbols (i.e. long-range dependencies). It would be good to see this empirically validated by evaluating performance on long sentences specifically. Minor comments: - Are you using dropout on the source/target embeddings? - Line 146: There seems to be dangling "2"

Reviewer 3



Summary: This paper presents an approach for machine translation using attention based layers. The model does not include convolution or rnns and still achieves state of the art on WMT14 English-German and English-French data sets. The model uses parallel attention layers whose outputs are concatenated and then fed to a feed forward position-wise layer. Qualitative Assessment: The paper reads well and is easy to follow. The experimental setup is clear and provides enough details for replication. The paper provides many useful hints such as scaled dot product attention which improves gradient flow. A lot of content is presented and I hope to see a more in depth version.